# Bioactive Secoiridoids in Italian Extra-Virgin Olive Oils: Impact of Olive Plant Cultivars, Cultivation Regions and Processing

**DOI:** 10.3390/molecules26030743

**Published:** 2021-01-31

**Authors:** Ilario Losito, Ramona Abbattista, Cristina De Ceglie, Andrea Castellaneta, Cosima Damiana Calvano, Tommaso R.I. Cataldi

**Affiliations:** 1Dipartimento di Chimica, Università degli Studi di Bari “Aldo Moro”, via Orabona 4, 70126 Bari, Italy; ramona.abbattista@gmail.com (R.A.); cristinadeceglie@libero.it (C.D.C.); a.castellanet@libero.it (A.C.); tommaso.cataldi@uniba.it (T.R.I.C.); 2Centro Interdipartimentale SMART, Università degli Studi di Bari “Aldo Moro”, via Orabona 4, 70126 Bari, Italy; cosimadamiana.calvano@uniba.it; 3Dipartimento di Farmacia-Scienze del Farmaco, Università degli Studi di Bari “Aldo Moro”, via Orabona 4, 70126 Bari, Italy

**Keywords:** secoiridoids, extra-virgin olive oil, horizontal centrifugation, olive cultivars, liquid chromatography-high resolution mass spectrometry

## Abstract

In the last two decades, phenolic compounds occurring in olive oils known as *secoiridoids* have attracted a great interest for their bioactivity. Four major olive oil secoiridoids, i.e., oleuropein and ligstroside aglycones, oleacin and oleocanthal, were previously characterized in our laboratory using reversed-phase liquid chromatography with electrospray ionization-Fourier transform-mass spectrometry (RPLC-ESI-FTMS). The same analytical approach, followed by multivariate statistical analysis (i.e., Principal Component Analysis), was applied here to a set of 60 Italian extra-virgin olive oils (EVOO). The aim was to assess the secoiridoid contents as a function of olive cultivars, place of cultivation (i.e., different Italian regions) and olive oil processing, in particular two- vs. three-phase horizontal centrifugation. As expected, higher secoiridoid contents were generally found in olive oils produced by two-phase horizontal centrifugation. Moreover, some region/cultivar-related trends were evidenced, as oleuropein and ligstroside aglycones prevailed in olive oils produced in Apulia (Southern Italy), whereas the contents of oleacin and oleocanthal were relatively higher in EVOO produced in Central Italy (Tuscany, Lazio and Umbria). A lower content of all the four secoiridoids was generally found in EVOO produced in Sicily (Southern Italy) due to the intrinsic low abundance of these bioactive compounds in cultivars typical of that region.

## 1. Introduction

With an average worldwide production approaching 3 million metric tons over the last five years and involving more than 40 different countries, according to statistical data provided by the International Olive Oil Council (IOC) [1], olive oil has become an internationally appreciated food product, well beyond the borders of its cradle, the Mediterranean Sea area. This success is related to the increasing spread of knowledge about the benefits that its regular consumption may have for human health, in turn related to the presence of several bioactive constituents, which have also led to consider olive oil as a functional food [2,3,4].

Although phenolic compounds occur in relatively low amounts in olive drupes and leaves, compared to other constituents, once transferred into olive oil they play an important role in determining the beneficial effects for human health [5,6,7,8,9]. Among them, special attention has been dedicated to compounds including a phenolic moiety but belonging, at the same time, to the class of secoiridoids. Like other plants of the Oleaceae family, olive (*Olea europaea* L.) is able to synthesize two major secoiridoids, oleuropein and ligstroside, corresponding to esters formed, respectively, by tyrosol (p-hydroxy-phenylethyl alcohol, HPEA) or 3-hydroxy-tyrosol (3,4-dihydroxy-phenylethyl alcohol, 3,4-DHPEA) and a secoiridoid corresponding to the glycosidic derivative of a carboxylic acid known as elenolic acid (IUPAC name 2-[(2S,3S,4S)-3-formyl-5-methoxycarbonyl-2-methyl-3,4-dihydro-2H-pyran-4-yl]acetic acid, usually abbreviated as EA) [2,3,4]. As a consequence of enzymatic and chemical reactions occurring at different stages of olive oil production, oleuropein and ligstroside aglycones, often abbreviated as 3,4-DHPEA-EA and HPEA-EA, respectively, and oleac(e)in and oleocanthal, including a decarboxymethylated form of elenolic acid (EDA) and thus usually abbreviated as 3,4-DHPEA-EDA and HPEA-EDA, respectively, arise from oleuropein and ligstroside, according to the case. The nutraceutical properties of these compounds, with a primary focus on their antioxidant activity, have been extensively studied for almost two decades, along with their influence on organoleptic features (e.g., pungency) and on the quality of olive oil [3,4,5,6,7,8,9,10,11,12,13,14]. During the same period, an extensive use of techniques like UV and NMR spectroscopies and then mass spectrometry, usually coupled to liquid chromatography, has led scientists to progressively unveil the peculiar structural versatility characterizing major olive oil secoiridoids [15,16,17,18,19,20,21,22,23,24,25,26,27,28,29].

Recently, the systematic analysis of extra-virgin olive oils (EVOO) by reverse phase liquid chromatography coupled to electrospray ionization with high resolution Fourier transform mass spectrometry (RPLC-ESI-FTMS) has enabled the recognition of the complex battery of isomers related to already known open*-* and closed-structure forms of olive oil major secoiridoids. Indeed, the presence of diastereoisomers, related to the presence of chiral centers, of positional/geometrical isomers, related to one of their C=C bonds (see Figure 1), and even of stable keto-enolic tautomers arising from the presence of enolizable C=O groups has been assessed [30,31,32,33], thus explaining the multiplicity of chromatographic peaks often detected for secoiridoids. The same analytical approach has also proved useful to study the oxidative and hydrolytic deterioration of major secoiridoids in an olive oil stored for up to six months under different conditions [32]. As a result, the non-negligible occurrence of oxidation of one of the carbonylic groups usually present on the structure of olive oil secoiridoids to carboxylic acid has been evidenced for oleuropein aglycone and for oleacin and oleocanthal [32].

In a parallel study, RPLC-ESI-FTMS, integrated by chemometric analysis, was applied to monitor major secoiridoids in a set of olive oils produced in the region of Apulia (Southern Italy) mainly using the same olive cultivar (*Coratina*) but different processing technologies [33]. As a result, horizontal centrifugation, exploited to separate oil from olive paste obtained after malaxation, was found to play a relevant effect on the final amount of major secoiridoids. Specifically, the two-phase procedure, i.e., the one not involving addition of water to the malaxed olive paste, was found to lead to a higher amount compared to the three-phase centrifugation process. This outcome was in accordance with those of previous works, that usually compared the effects of the two types of horizontal centrifugation on olive oils produced under controlled conditions (i.e., by fixing other technological parameters) [34,35,36]. The result is due to the fact that the introduction of additional water into the horizontal decanter favors the transfer of secoiridoids into the aqueous fraction during centrifugation, thus reducing their final amount in olive oil. In the same study, the adoption of vertical centrifugation at the end of the production process, for the separation of residual water from olive oil, was found to decrease the content of oleuropein and ligstroside aglycone [33]. This finding was in accordance with a previous investigation, showing that the general decrease occurring in the secoiridoid amount in olive oils resulting from vertical centrifugation was related to an increased oxidative degradation involving secoiridoids during this process, compared to mild sedimentation [37].

The final secoiridoid content in olive oil is also significantly affected by factors like olive cultivar, geographical origin, and cultivation practices. Indeed, biosynthetic pathways leading to the precursors of olive oil secoiridoids, i.e., oleuropein and ligstroside, can be significantly influenced by the expression of involved enzymes, in turn related to cultivar-specific genetic factors (see Ref. [38] and references cited therein). The effect of geographical origin can be expressed as a combination of the influence of climatic conditions and soil characteristics on the final amount of secoiridoid precursors in olive drupe, that, additionally, is influenced also by agronomical practices (such as irrigation, fertilization, ripening stage of drupes at the moment of collection, etc.) and even by the occurrence of pests and diseases [38].

Starting from this background, a systematic investigation was undertaken in our laboratory on 60 EVOO produced in five different Italian regions during two campaigns (2016/2017 and 2017/2018), with the goal of assessing if, and to what extent, the final amount of secoiridoids was affected by olive cultivars, geographical origin, and processing factors. The map reported in Figure 1 indicates which regions were considered: Apulia and Sicily for Southern Italy and Tuscany-Lazio-Umbria for Central Italy. As evidenced by the percentage production data averaged over the 2016–2019 period (source: ISMEA, Institute of Services for Agricultural and Food Market [39]), reported with the map, these areas have variable relevance in terms of national EVOO production, with Apulia representing the leading region, among those selected. Moreover, as emphasized in Table 1, where detailed information available for each sample is reported, they enabled the considerations of cultivars specific for each geographic origin, for which preliminary data on the secoiridoid content were also available [34,40,41] and could thus be used for comparison purposes. As for climatic aspects, most of Apulia and Sicily territories are classified as hot temperate climate zones (Csa, according to the Koppen climate classification), with a narrow subtropical temperate (Cs) zone along the coastline in the case of Sicily [42]. On the other hand, excepting the coastline of Tuscany and Lazio (representing hot temperate climate zones), the regions of Central Italy considered for the EVOO selection are characterized by a sub-coastal temperate climate, with generally lower temperatures, especially in summer, and a less pronounced water stress, compared to Apulia and Sicily [42].

The main results of the study, based on the elaboration of RPLC-ESI-FTMS data referred to the four major olive oil secoiridoids using Principal Components Analysis, will be described and discussed in the present paper.

## 2. Results

### 2.1. RPLC-ESI(-)-FTMS Analysis of Olive Oil Secoiridoids

As previously demonstrated [30,31,32,33], the RPLC-ESI(-)FTMS method recently developed in our laboratory enabled the systematic separation of several isoforms of the four major secoiridoids extracted from EVOO samples collected for the present investigation. Specifically, their occurrence was inferred from eXtracted Ion Current (XIC) chromatograms obtained from RPLC-ESI(-)-FTMS data using 0.0040 *m*/*z* unit wide intervals centered on exact ratios referred to secoiridoids [M-H]^-^ ions generated during the ESI process. An example of XIC traces retrieved for a specific EVOO extract (in this case referred to sample #82 reported in Table 1) is reported in Figure 2. By analogy with previous data [30,31,32,33], 13/14 different chromatographic peaks were usually detected for isoforms related to the aglycones of oleuropein (OA) and ligstroside (LA), with the former eluting respectively earlier than the latter, due to the increase in polarity induced by the presence of an additional phenolic OH group in its structure. Molecular structures reported in Figure 3 (each labelled with the numbers referred to the corresponding peaks in XIC traces of Figure 2) clarify the identity of each chromatographic feature, as inferred through a careful interpretation of MS/MS data and of H/D exchange experiments described in previous papers [30,31,32,33]. Isoforms with open (*Open Forms I* and *II*) or closed (*Closed Forms I* and *II*) structure were recognized for OA and LA. They were characterized by a combination of diastereoisomerism, due to carbon atoms labelled with an asterisk, with wavy bonds symbolizing both the possible steric configurations on all those atoms but the fixed chiral center on C^5^, geometric isomerism on the C^8^–C^9^ double bond (see the *E,Z* notation in Figure 3), and positional isomerism, due to the alternative location of this bond between C^8^ and C^10^. H/D exchange experiments enabled us to assess that stable enolic (or even dienolic, in some cases) counterparts of aldehydic isoforms could also be present in EVOO extracts [30,31]. Moreover, the artificial cleavage of the glycosidic bond of oleuropein/ligstroside precursors, catalyzed by a commercial β-glucosidase, indicated *Closed Forms I* to be the initial derivatives of cyclic hemiacetals arising from the detachment of glucose from oleuropein/ligstroside. Subsequent incubation at acid pH led to cycle opening to provide *Open Forms I* and *II* and, through a complex re-cyclization process, *Closed Forms II* as well [30,31]. Therefore, it was not surprising that the profiles found for OA/LA isoforms after the systematic analysis of the EVOO samples considered during this study could vary even significantly, depending on specific conditions (endogenous enzyme availability, pH, temperature, etc.) occurring during production and even on intrinsic differences in the amount of their glycosidic precursors (oleuropein and ligstroside) in olive drupes used for oil production.

As shown by the corresponding XIC traces in Figure 2, the scenario was much simpler for oleacin and oleocanthal, likely because their structures miss a stereogenic center on C^4^, due to the lack of the carboxymethyl (COOCH_3_) moiety, and, additionally, this feature prevents *Closed Forms II* from being formed. In this case both open-structure diastereoisomers related to isoforms including a C^8^–C^10^ double bond (*Open Forms II* in Figure 3) prevailed in XIC traces, whereas their positional isomers with a C^8^-C^9^ double bond (*Open Forms I*) and the closed structure isoforms (*Closed Forms I*) had a very low, if any, relevance [32,33]. It was thus not surprising that they were almost absent in the case of oleacin in Figure 2, and that the lately eluting peak in the corresponding XIC trace (r.t. 15.61 min) was actually an oxidized derivative of oleocanthal. Namely, this is a derivative with a C=O group turned into a COOH one, often called *oleocanthalic acid*, recognized only through a peculiar fragment detected in its MS/MS spectrum, since it is perfectly isobaric with oleacin (the two compounds have the same chemical formula) [32]. Notably, a closed structure isoform is reported in Figure 3 for this derivative since it was proved to be the first carboxylic by-product generated from oleocanthal; yet, after long storage times (3–6 months) with periodical exposure to oxygen, also open-structure isoforms of this secoiridoid were found to undergo oxidative deterioration [32].

It is also worth noting that the generation of carboxylic by-products was common to all the four major secoiridoids of EVOO but ligstroside aglycone, which appeared to resist to oxidative deterioration even after 6 months of storage with deliberate periodic exposure to atmospheric oxygen [32]. This was obviously not the type of storage adopted for olive oils included in the present study, since their bottles were kept closed until analysis. On the other hand, since RPLC-ESI(-)-FTMS analysis had to be performed after some months of storage for some samples, due to practical reasons, the occurrence of oxidized by-products, related to the presence of oxygen originally dissolved into the olive oil matrix [32], could not be completely excluded and had to be carefully taken into account before deciding how to calculate the values of variables adopted for Principal Component Analysis (vide infra).

A final consideration about data shown in Figure 2 is deserved by the XIC trace referred to oleuropein. This glycosidic secoiridoid was never detected in EVOO extracts, which was reasonable, since oleuropein contained in drupes is expected to be completely converted into its aglycone during olive oil production. For this reason, it could be used as an internal standard, added at a 100 mg/L concentration to each olive oil extract. The peak area inferred from its XIC chromatogram could thus be adopted to normalize those resulting from the integration of all features appearing in the XIC traces of the four EVOO secoiridoids. This procedure enabled the minimization of the effects of instrumental response fluctuations on absolute peak areas, that could not be excluded over the time range required to analyze the large number of samples considered during this study.

### 2.2. Principal Components Analysis Based on RPC-ESI(-)-FTMS Responses of the Four Major Secoiridoids in 60 Italian EVOO

In a preliminary step of the chemometric investigation, RPC-ESI(-)-FTMS data (normalized XIC peak areas) referred to OA, LA, oleacin and oleocanthal and to their oxidized derivatives (carboxylic acids), except LA (see previous section), were considered in the perspective of PCA-based elaborations, as already described with Apulian EVOO samples [33]. However, in this case the incidence of oxidized derivatives could be influenced by the time elapsed between production and analysis, which was not necessarily related to the region or the technology of production of a specific sample. As a result, an artificial increase in the variability perceived in PCA scatterplots. On the other hand, ignoring the presence of oxidized derivatives might have led to underestimate the amount of the corresponding precursors, if the incidence of oxidation was not negligible, thus potentially altering the original distribution of secoiridoids in a specific sample. As a compromise between the issues now described, normalized peak areas obtained for oxidized derivatives were summed to those of the respective precursors and such combined variables (with the exclusion of LA) were considered for PCA-based elaborations, thus reducing to four the total number of variables. It is worth noting that oxidized derivatives share one of their ionization sites (a phenolic OH group) with their precursors, although they include a COOH group in their molecular structure, which is expected to improve the negative ionization yield during the ESI process, compared to that of the corresponding precursors.

Each EVOO sample was subjected to two replicated extractions and subsequent LC-MS analyses, with extracts spiked with oleuropein 100 mg/L, adopted as internal standard, thus average values of normalized peak areas obtained from each replicate were adopted as variables for PCA. Moreover, the *correlation matrix* method was adopted for PCA and, as a first approach, the clustering of EVOO samples based on the type of horizontal centrifugation was evaluated, as done in our previous paper on Apulian olive oils [33]. The scatter and the loading plot obtained at the end of PCA elaboration for the first two principal components, accounting for 85.6% of the total variance, are thus shown in Figure 4 for olive oils of the 2016/2017 campaign, with samples labelled with a different color according to the type of horizontal centrifugation (red—two phase; blue—three phase). Despite the variable geographical origin of EVOO samples under consideration, a distinction based on the type of horizontal centrifugation could be observed, and it was consistent with the one inferred for Apulian oils in our previous paper [33]. Indeed, a general increase in the normalized response for all the four major secoiridoids was observed when passing from three- to two-phase horizontal centrifugation, in accordance with literature [34,35,36]. Moreover, also in this case the presence of a similar trend in terms of loadings for oleocanthal and oleacin, on one side, and for the aglycones of oleuropein and ligstroside, on the other one, was observed.

Borderline samples emerged from a careful evaluation of the scatter plot shown in Figure 4. In particular, sample #4, produced in Apulia using three-phase horizontal centrifugation and *Coratina* olives (see Table 1), appeared more concentrated in the four secoiridoids, especially OA and LA, then it would have been expected for that type of centrifugation. A possible explanation for this outcome could be the well-known high secoiridoid content typical of the *Coratina* cultivar [34]. Another oil obtained using three-phase centrifugation, sample #21, produced in Lazio with a blend of cultivars, appeared richer in secoiridoids, especially oleacin and oleocanthal, then expected. Unfortunately, the cultivars adopted in the blend of olives adopted for its production could not be declared by the producer, thus a further evaluation was not possible. On the opposite side, amounts slightly lower than expected were inferred for secoiridoids in samples #17, 22 and 24, all obtained using two-phase horizontal centrifugation (see Figure 4). The outcome for sample #17 was particularly surprising, since it was produced from secoiridoid-rich *Coratina* olives. However, vertical centrifugation was used by the producer in the last step of oil production and, based on considerations made before, it could have played a role in reducing the final amount of secoiridoids. The same occurred for samples #22 and 24; in the former case olive cultivars adopted in the blend were not disclosed by the producer, whereas a declared blend of four cultivars was adopted for oil #24. According to the literature, three of them, *Frantoio*, *Moraiolo* and *Leccino*, are generally less abundant in secoiridoids, compared, for example, to *Coratina* [34], thus the result for sample #24 was not completely surprising.

To search for correlations between secoiridoid content and geographical origin, the set of 26 EVOO samples of the 2016/2017 campaign was divided into two groups (13 samples each), based on the type of horizontal centrifugation adopted during their production. Afterwards, PCA was performed separately on the two groups and the resulting scatter and loading plots are reported in Figure 5 and Figure 6 for olive oils obtained using a three-phase or a two-phase decanter, respectively. Note that ellipses were drawn (using colors consistent with those related to each region in the map shown in Figure 1) in all PCA scatter plots reported in this work just to emphasize the relationship of samples with specific regions. As shown by Figure 5, Apulian EVOO samples were located on the left side of the scatter plot, due to the incidence of the aglycones of oleuropein and ligstroside (see the loading plot in the same figure), although this feature was more relevant for sample #4, produced using *Coratina* olives, in accordance with the results shown in Figure 4. Interestingly, secoiridoids were less abundant in Apulian oils produced using *Frantoio* or *Ogliarola* olives, in accordance with previous studies [11,34,36]. Among the considered samples, Sicilian olive oils appeared as those exhibiting the lowest amounts of the four secoiridoids. This finding was in excellent agreement with data already reported for EVOO produced with the two cultivars involved, i.e., *Nocellara* and *Tonda Iblea* [40], with the second one being characterized by a lower content compared to the first, as also suggested by the scatter plot reported in Figure 5. The combination of the inherently low content in secoiridoids of the employed cultivars and of the three-phase horizontal centrifugation, that usually leads to a partial loss of these compounds, is thus able to explain the peculiar position of Sicilian oils in the plot. As for the five EVOO produced in Lazio or Tuscany, the location of their points in the plot was generally consistent with the common geographical area (Central Italy), yet the two samples from Lazio obtained using blends of cultivars (undisclosed by the producers) appeared to be distinct from the others due to a characteristic abundance in oleacin and oleocanthal. As for the two olive oils from Tuscany, i.e., samples #23 and 15, the respective location in the plot was consistent with the lower abundance in secoiridoids reported for the *Olivastra seggianese* cultivar [41] compared to the *Moraiolo* one [11], which was one of the cultivars used to produce EVOO sample #15, in combination with *Frantoio* and *Leccino*, which, according to data reported in Refs. [11,41], should contain similar amounts of major secoiridoids.

Turning to olive oils produced during the 2016/2017 campaign using two-phase horizontal centrifugation, the corresponding scatter plot, shown in Figure 6, confirmed the good separation between Apulian oils, all obtained using the *Coratina* cultivar in this case, and those produced in Central Italy (Sicilian olive oils produced using a two-phase decanter were not available in this set). Based on the loading plot, the position of Apulian oils in the scatter plot was related to a peculiar abundance in ligstroside aglycone, along with that of oleuropein aglycone, as already observed. On the other hand, oils produced in Central Italy appeared to be divided into two clusters: the two samples from Lazio were located in the upper-right part of the scatter plot, as a consequence of the abundance of oleocanthal, whereas those from Tuscany and Umbria were located at negative values of the second principal component. Notably, both Lazio olive oils were not subjected to vertical centrifugation at the end of the productive process; this feature seems to have influenced positively their oleocanthal content. A further interesting issue was that just Tuscany olive oils obtained from olive blends not disclosed by producers (samples #5, 14 and 22) were related to negative values of the first principal component, a feature influenced by the concentration of OA, whereas other samples, including two from Umbria, involving typical cultivars of Central Italy (*Moraiolo*, *Frantoio*, *Leccino*), were located at increasing values of that component, where oleacin and oleocanthal have more influence among variables. Thus, once the horizontal centrifugation type was fixed, some geographical trends emerged (see Figure 5 and Figure 6) mainly as a consequence of the use of cultivars specific of each area, with Apulian oils resulting well distinct from others due to the prevailing use of *Coratina*, leading to relevant concentrations in OA and LA. On the other hand, EVOO from Central Italy were generally characterized by a higher relative abundance in oleocanthal and/or oleacin, excepting those produced in Tuscany employing two-phase horizontal centrifugation and undisclosed blends of olives (see the scatter plot in Figure 6). This outcome would lead to speculate that *Coratina* olives could have been involved, at some extent, in the adopted blends.

The effect of horizontal centrifugation and of region/cultivar was subsequently explored also on the 34 olive oils produced during the 2017/2018 campaign. The scatter and loading plots obtained from PCA for the entire set of olive oils, with sample points colored according to the type of horizontal centrifugation (red—two phase; blue—three phase) are reported in Figure 7. Compared to samples of the 2016/2017 campaign, a higher degree of mixing between olive oils produced with a different horizontal decanter was observed. In particular, at least 5 EVOO produced using a two-phase decanter (samples #42, #51, #81, #64 and #74) were found at low values of the first principal component, indicating a lower content of secoiridoids (see the loading plot in Figure 7), more similar to the one expected for oils produced with a three-phase decanter. As evidenced in Table 1, oils #42 and 51 were produced in Sicily using drupes of cultivars *Nocellara* (mixed with *Biancolilla*) and *Tonda Iblea*, that, as cited before, contain lower amount of secoiridoids [40]. In this case, even the use of a two-phase decanter was unable to raise the concentration of these compounds in the final oil. Samples #64 and #81 were both produced in Lazio and using *Leccino* as the main cultivar (mixed with *Carboncella* in one case), thus their outcome was surprising, since a relatively high concentration in oleocanthal and oleacin was expected, based on the results obtained for the 2016/2017 campaign. Furthermore, sample #74 was expected to contain high amounts of secoiridoids, especially OA and LA, since it was produced with *Coratina* olives; yet, according to the scatter plot shown in Figure 7, it appeared as an olive oil produced with a three-phase decanter. Interestingly, when further technological details were asked for to the respective producers, it emerged that EVOO #64, #74 and #81 were all subjected to press filtration with cardboard, a procedure which has been already reported to remove part of phenolic compounds from olive oils [43].

As for anomalous samples indicated by the PCA scatter plot of Figure 7 among those produced using a three-phase decanter, i.e., samples #56, #57, #58, #82 and #171, all located at values of the first principal component typical of oils produced with a two-phase decanter, the former three were produced from *Coratina* olives, and this could have influenced positively the final content in secoiridoids. The EVOO sample #82 was produced in Tuscany using a blend of olives whose cultivars were not disclosed by the producer. As discussed before, this finding might be related to the use also of secoiridoid-rich olives in the blend, e.g., *Coratina* ones, since the oil appeared to be relatively rich in OA and LA. An explanation for the relatively high abundance of secoiridoids in sample #171, produced in Umbria using a three-phase decanter and a blend of *Moraiolo*, *Frantoio* and *Leccino* olives, could not be found.

Additional evaluations on olive oils produced during the 2017/2018 campaign could be made after performing PCA on samples split in terms of type of horizontal centrifugation. As shown in Figure 8 for samples obtained using a three-phase decanter, the separation between Apulian and Sicilian olive oils in the scatter plot was, again, apparent and likely related to the intrinsic difference in secoiridoids abundance of the respective cultivars, i.e., *Coratina* for all Apulian oils, *Nocellara* for the Sicilian ones). As for olive oils produced in the Central Italy regions, all those from Lazio, obtained using blends including *Leccino* and *Frantoio* or *Carboncella* olives, were located in the upper-left quadrant of the score plot. On the other hand, already discussed samples #82, from Tuscany, and #171, from Umbria, were displaced towards Apulian samples, reflecting a higher-than-expected content in secoiridoids.

As for olive oils produced using a two-phase decanter, the scatter plot reported in Figure 9 showed, once again, the separation between Sicilian and Apulian samples. Among olive oils produced in Central Italy, the two from Lazio (samples #64 and #81) were separated because of the already discussed low abundance in secoiridoids. All other samples from Central Italy were in or close to the upper-right quadrant of the scatter plot, suggesting, once again, the peculiar incidence of oleocanthal and oleacin (see the loading plot reported in Figure 9). Interestingly, oils #77 and 80 from Tuscany appeared closer to the Apulian ones, which indicated a higher-than-expected incidence of OA and LA, at least considering the cultivars adopted for their production (*Moraiolo* as such or in blend with *Frantoio* and *Leccino*). The use of a blade crusher, instead of the more typical hammer one, in the first step of production (see Table 1) might have played a role in the final content of secoiridoids of these two samples. Indeed, according to a previous study, the blade crusher might reduce the release of endogenous oxidoreductases from the drupe stone, thus limiting oxidative processes occurring on phenolic compounds, including secoiridoids. This would ultimately lead to a decrease of their concentration in the final product, especially if severely oxidized by-products are removed from olive paste through volatilization [44].

## 3. Discussion

The systematic RPLC-ESI-FTMS analysis of the four major secoiridoids, i.e., oleuropein and ligstroside aglycones, oleacin and oleocanthal, in 60 commercial EVOO produced during two campaigns (2016/2017 and 2017/2018) in different regions of Italy and using a variety of cultivars and technological approaches gave the opportunity to evaluate how these factors may affect the final content of these important bioactive compounds in olive oil when real conditions of production are considered. Following a previous study performed in our laboratory on a smaller set (22 samples) of nearly mono-cultivar (*Coratina*) olive oils produced in Apulia [33], the effect of horizontal centrifugation was first evaluated by PCA on each of the two groups of samples related to production year. The enhancing effect played by two-phase horizontal centrifugation in terms of final secoiridoid content, due to a limited loss of secoiridoids into vegetation waters compared to three-phase centrifugation, was generally confirmed by PCA, especially in the case of EVOO produced during the 2016/2017 campaign. This outcome was confirmed by comparing 95% confidence intervals for average values of normalized peak areas of secoiridoids obtained from EVOO produced during that campaign with a three- (3P) or a two- (2P) phase horizontal centrifugation. In particular, the following values were obtained for the 3P vs. 2P comparisons: OA) 1.5 ± 0.8 vs. 4.5 ± 0.9; LA) 1.0 ± 0.6 vs. 2.4 ± 0.7; Oleacin) 0.8 ± 0.3 vs. 1.4 ± 0.2; Oleocanthal) 0.53 ± 0.15 vs. 0.75 ± 0.15. The application of the Tukey test showed a statistically significant increase, at a 5% significance, for each secoiridoid in EVOO produced using a two-phase horizontal decanter. Not surprisingly, based on the PCA outcome shown in Figure 7, this was not the case when EVOO samples produced during the 2017/2018 campaign were considered. Indeed, the following confidence intervals were obtained (3P vs. 2P): OA) 4.3 ± 1.2 vs. 5.0 ± 1.2; LA) 2.7 ± 1.0 vs. 2.9 ± 0.9; Oleacin) 1.4 ± 0.3 vs. 1.4 ± 0.4; Oleocanthal) 0.8 ± 0.2 vs. 0.9 ± 0.2, thus a significant difference could not be found for any secoiridoid. As explained before, when interpreting the PCA outcomes for EVOO produced during the 2017/2018 campaign, this result was related to the presence in the dataset of olive oils produced using three-phase horizontal centrifugation and drupes of a cultivar quite rich in secoiridoids, like *Coratina* from Apulia, that was able to enhance remarkably their final content in the oils despite the loss occurring during three-phase centrifugation. The opposite effect was observed, in the same dataset, for olive oils produced by two-phase centrifugation but using drupes of cultivars containing a lower amount of secoiridoids, like those typical of the Sicily region. The presence of these peculiar samples contributed to make the average secoiridoid contents in the two datasets similar.

Interesting regional/cultivar trends could be observed in the PCA scatter plots when samples sharing a specific type of horizontal centrifugation were considered. Indeed, olive oils were found to be generally clustered according to one of the following regions or geographical areas: Apulia, Sicily (both located in Southern Italy) and Central Italy (including, in this case, the regions of Tuscany, Lazio and Umbria). Apulian EVOO had their specificity in the relevant amount of oleuropein and ligstroside aglycones, whereas oils from Central Italy were peculiar for the relative high amount of oleacin and, at a minor extent, oleocanthal. Regardless of the type of horizontal centrifugation adopted, all four main secoiridoids were generally found at their lowest concentrations in olive oils produced in Sicily. Based on the information previously reported on the secoiridoid content in drupes/leaves of specific Italian cultivars, these trends were found to reflect the use of region-specific olive cultivars, i.e., *Coratina* in Apulia, *Leccino*, *Moraiolo* and *Frantoio* in Central Italy and *Nocellara* in Sicily. Nonetheless, some olive oils produced in Tuscany using blends of cultivars undisclosed by producers for confidentiality reasons appeared to have an intriguing similarity with Apulian oils in terms of oleuropein and ligstroside aglycones content, thus leading to hypothesize that a mixture of local cultivars with the secoiridoid-rich *Coratina* might have been used to optimize the characteristics of the final product.

## 4. Materials and Methods

### 4.1. Chemicals and Extra-Virgin Olive Oil Samples

The following chemicals: water, methanol and acetonitrile (LC-MS grade), n-hexane (HPLC-grade), and oleuropein (IUPAC name: (4S,5E,6S)-4-{2-[2-(3,4-dihydroxyphenyl)ethoxy]-2-oxoethyl}-5-ethylidene-6-{[(2S,3R,4S,5S,6R)-3,4,5-trihydroxy-6-(hydroxymethyl)-2-tetrahydropyranyl]oxy}-4H-pyran-3-carboxylic acid methyl ester), were purchased from Sigma-Aldrich (Milan, Italy).

Then, 60 extra-virgin olive oils produced during the 2016/2017 (26) and the 2017/2018 (34) campaign and kindly gifted by the respective producers were obtained in the context of a national research project on EVOO (Project VIOLIN: Valorization of Italian OLive products through INnovative analytical techniques). Each producer was asked to provide, on a voluntary basis, relevant production information on the corresponding olive oil, which is summarized in Table 1. As described before, the same number of samples (30) were obtained using a two-phase or a three-phase horizontal centrifugation as part of the production process and the following five Italian regions were involved: Apulia and Sicily (Southern Italy), Lazio, Tuscany and Umbria (Central Italy). As emphasized in Table 1, several different olive cultivars were used, singly or mixed in a blend, to produce the analyzed oils; only occasionally, cultivars adopted in a blend were not disclosed by the producer for confidentiality reasons. Each of the selected oils was divided into 50 mL aliquots in the laboratories of the leading research unit of the VIOLIN project. Afterwards, each aliquot, stored in a dark glass small bottle with screw cap, was delivered to other research units, including ours, where each bottle was stored in the dark at room temperature and opened just before sampling the amount of olive oil required for the extractions of secoiridoids.

### 4.2. Extraction of Secoiridoids from Extra-Virgin Olive Oils

Secoiridoids were extracted in duplicate from the selected EVOO samples, along with other polar constituents, using a CH_3_OH/H_2_O 60:40 (*v*/*v*) mixture, according to a protocol already adopted in our laboratory [30,31,32,33], in turn adapted from those reported previously by Vichi et al. [23] and Ricciutelli et al. [28]. In particular, each of two 2 g aliquots of extra virgin olive oil were dissolved into 3 mL of HPLC-grade hexane and vortexed for 1 min, then 500 μL of the extracting solvent mixture were added. The new mixture was vortexed for 2 min and then sonicated for 4 min using a DU-32 ultrasonic bath (Argo Lab, Carpi, Italy), operated at 40 kHz frequency, 120 W power and 23 °C temperature. Centrifugation at 2000× *g* for 5 min was performed to achieve the separation of the hexane-rich phase from the methanolic-aqueous one, the one including secoiridoids, that was carefully withdrawn with a microsyringe and stored in a glass tube with the headspace saturated with nitrogen. The hexane-rich phase was subjected to a further extraction with 500 μL of the extracting solvent mixture, to extract eventual residual secoiridoids. The two aliquots of methanolic-aqueous extract were finally pooled, washed for 1 min with 2 mL of n-hexane under vortexing, to remove eventual residual apolar compounds, and then centrifuged for 5 min at 2000× *g*, to separate the methanolic-aqueous phase. The latter was subsequently stored at +4 °C in a glass vial, closed with a screw cap, whose headspace was saturated with nitrogen to minimize the eventual oxidation of extracted secoiridoids before RPLC-ESI-FTMS analysis. As discussed in our recent papers [32,33], the time of exposure to methanol at room temperature, during the extraction step, was short enough to minimize the transformation of extracted secoiridoids into methanol-involving hemiacetals/acetals. Indeed, the incidence of these by-products was found to be lower than 5%, based on their mass spectrometric responses.

### 4.3. RPLC-ESI-FTMS Instrumentation and Operating Conditions

RP LC-ESI-FTMS separations on olive oil extracts were performed using an Ultimate 3000 UHPLC system coupled to a Q-Exactive quadrupole-Orbitrap mass spectrometer (Thermo Scientific, Waltham, MA, USA). RPLC separations were performed using an Ascentis Xpress C18 column (150 × 2.1 mm ID, 2.7 µm particle size) preceded by an Ascentis Xpress C18 (5 × 2.1 mm ID) security guard cartridge (Supelco). 5 μL of olive oil extracts were injected using the autosampler of the Ultimate 3000 system. Before injection each extract was spiked with oleuropein 100 mg/L, used as internal standard, i.e., to normalize mass spectrometric responses referred to target secoiridoids to that of oleuropein, which was always absent from the oil extracts. As shown in our recent paper [32], this normalization compensated for fluctuations in the instrumental response and contributed to the good reproducibility observed for replicated extracts.

RPLC separations of secoiridoids were performed using the following elution gradient, based on water (solvent A) and acetonitrile (solvent B), already adopted in our laboratory for the separation of isomeric secoiridoids in olive oil extracts [30,31,32,33]: 0–5 min) 20% solvent B; 5–35 min) from 20% to 50% (*v*/*v*) solvent B; 35–40 min) from 50% to 100% of solvent B; 40–50 min) isocratic at 100% solvent B; 50–55 min) from 100% to 20% of solvent B; 55–70 min) column reconditioning at 20 % solvent B. The flow rate was always set at 200 μL/min and the column temperature at 25 °C. All major secoiridoids and their eventual oxidative by-products were easily detected as [M-H]^−^ ions, because of deprotonation occurring during the ESI process, involving a phenolic OH group or a COOH group, according to the case, thus negative polarity was always adopted for MS detection. ESI(-)-FTMS full scan acquisitions were performed in the *m*/*z* range 100–1500 after setting the main parameters of the heated ESI (HESI) interface and of the ion optics of the Q-Exactive spectrometer as follows: sheath gas flow rate, 60 (arbitrary units); auxiliary gas flow rate, 15 (arbitrary units); spray voltage, −4 kV; capillary temperature, 200 °C; S-lens RF level, 100 (arbitrary units). MS acquisitions were performed by setting the resolving power of the Q-Exactive spectrometer at its maximum (120,000 at *m*/*z* 200), which resulted in a resolving power always better than 110,000 for signals related to all analytes (note that the resolving power of an Orbitrap mass analyzer decreases at the increase of the *m*/*z* ratio). A mass accuracy always better than 2 ppm was achieved with the Q-Exactive spectrometer, thanks to a daily calibration, performed before starting LC-MS analyses, based on infusion-ESI(-)-FTMS analysis of the PierceTM Negative Ion Calibration Solution (sodium dodecyl sulfate 2.9 µg/mL, sodium taurocholate 5.4 µg/mL and 0.001% Ultramark 1621), as recommended by the spectrometer manufacturer.

The LC-MS instrumentation was controlled by the Xcalibur software (Thermo Scientific), used also for ion current extraction required to obtain the mass spectrometric response for each secoiridoid of interest.

### 4.4. Chemometrics

RPLC-ESI(-)-FTMS data obtained from two extracts of each of the 60 Italian olive oils object of this study were elaborated using Principal Components Analysis (PCA). Peak areas obtained from XIC traces referred to the four secoiridoids of interest were used as variables for PCA elaborations, after being normalized to the peak area referred to oleuropein 100 mg/L, added to each sample as internal standard. Specifically, normalized peak areas of ligstroside aglycone were used as such, since no significant oxidation of this secoiridoid was observed in EVOO extracts, in accordance with results discussed in our recent papers [32,33]. Conversely, oleuropein-normalized XIC peak areas of oleuropein aglycone, oleacin and oleocanthal were summed, respectively, to those found for the corresponding carboxylic acids, since the incidence of these oxidative by-products was not negligible for those secoiridoids (see Section 2.2). Normalized responses obtained for each secoiridoid after the analysis of the two extracts of each EVOO were averaged before being used as the PCA input value for that compound in the selected sample. An outline of the described process has been added to the Appendix A. PCA was performed using the Minitab^®^ 18.1 software (Minitab LLC, State College, PA, USA), after selecting the *correlation matrix* approach.

## 5. Conclusions

The integration between RPLC-ESI(-)-FTMS analysis and Principal Component Analysis (PCA) was confirmed to be a powerful tool to study major secoiridoids in extra-virgin olive oil extracts, giving the opportunity to investigate the eventual influence of processing factors, olive cultivar and geographical origin on an extended set of oils, produced in 5 different Italian regions during two subsequent campaigns.

A confirmation of the important role played by horizontal centrifugation, among processing factors, in determining the final content of bioactive secoiridoids in extra-virgin olive oils was obtained, with two-phase centrifugation leading to generally higher final concentrations. On the other hand, the cultivar-specific content of olive oil secoiridoid precursors, i.e., oleuropein and ligstroside, was found to exert a non-negligible effect, sometimes subverting the one related to horizontal centrifugation. When this factor was fixed, the natural content of precursors in olive drupes contributed to determine peculiar regional trends of the olive oils secoiridoids, since region-specific cultivars were usually adopted for production, following a typical practice in Italy. Nonetheless, when some olive oils produced from undisclosed blends of cultivars were analyzed, hints of the presence of a cultivar typical of a region different from the one of production were obtained from PCA.

## Figures and Tables

**Figure 1 molecules-26-00743-f001:**
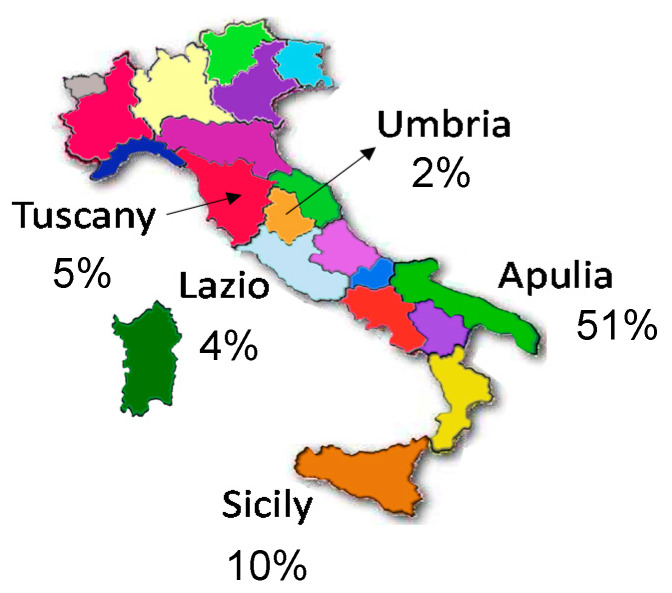
A map of Italy evidencing regions in which the 60 EVOO considered in the present study were collected. The 2016–2019 percentage averages of production are reported for each region (source: ISMEA, Institute of Services for Agricultural and Food Market [39]).

**Figure 2 molecules-26-00743-f002:**
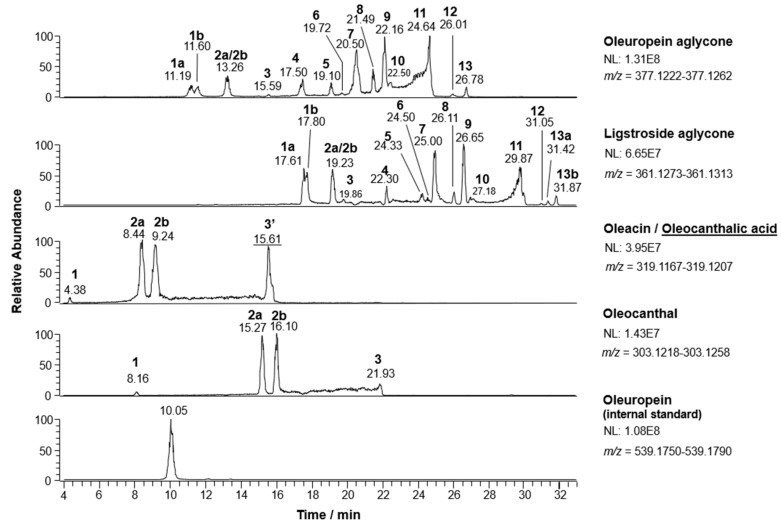
eXtracted Ion Current (XIC) chromatograms obtained for major secoiridoids and for internal standard oleuropein (100 mg/L) after the RPLC-ESI(-)-FTMS analysis of the methanol/water (60/40 *v*/*v*) extract of an extra-virgin olive oil (sample #82 in Table 1).

**Figure 3 molecules-26-00743-f003:**
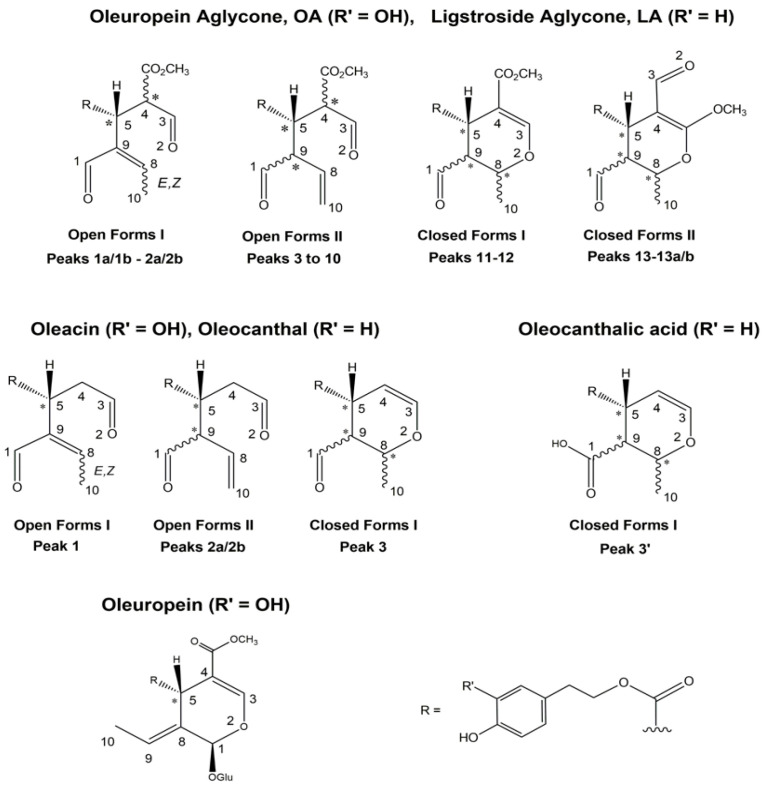
Molecular structures referred to different chromatographic peaks detected in XIC traces of major secoiridoids (see Figure 2), based on the characterization described in Refs. [30,31,32,33]. Asterisks indicate stereogenic centers (variable configuration at C^8^ and/or C^9^ is emphasized by a wavy bond to the methyl group); the *E*,*Z* notation is referred to the geometry of the C^8^–C^9^ double bond, emphasized by the drawing of a wavy bond between C^8^ and C^10^. For the sake of simplicity, enolic/dienolic counterparts, also identified in extracts, were not included in the figure. Note that the numbering of atoms conventionally adopted for secoiridoids in the literature has been used.

**Figure 4 molecules-26-00743-f004:**
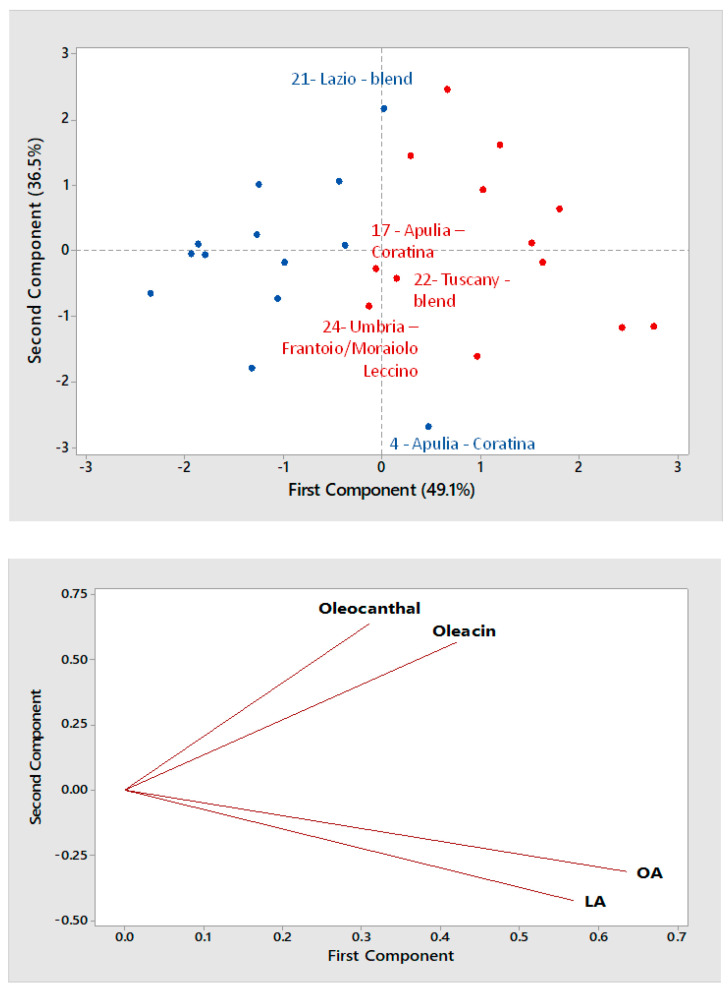
Score plot (**top panel**) and Loading plot (**bottom panel**) referred to the first two components provided by Principal Components Analysis (PCA) based on average normalized peak areas obtained for OA, LA, oleacin and oleocanthal (including the contribution of their carboxylic derivatives, if present) after the RPLC-ESI(-)-FTMS analysis of two replicated extracts for each of 26 Italian EVOOs produced during the 2016/2017 campaign. Olive oils produced using a three- or a two-phase horizontal decanter are indicated by blue and red dots, respectively. The sample numbering for emphasized samples is the same adopted in Table 1.

**Figure 5 molecules-26-00743-f005:**
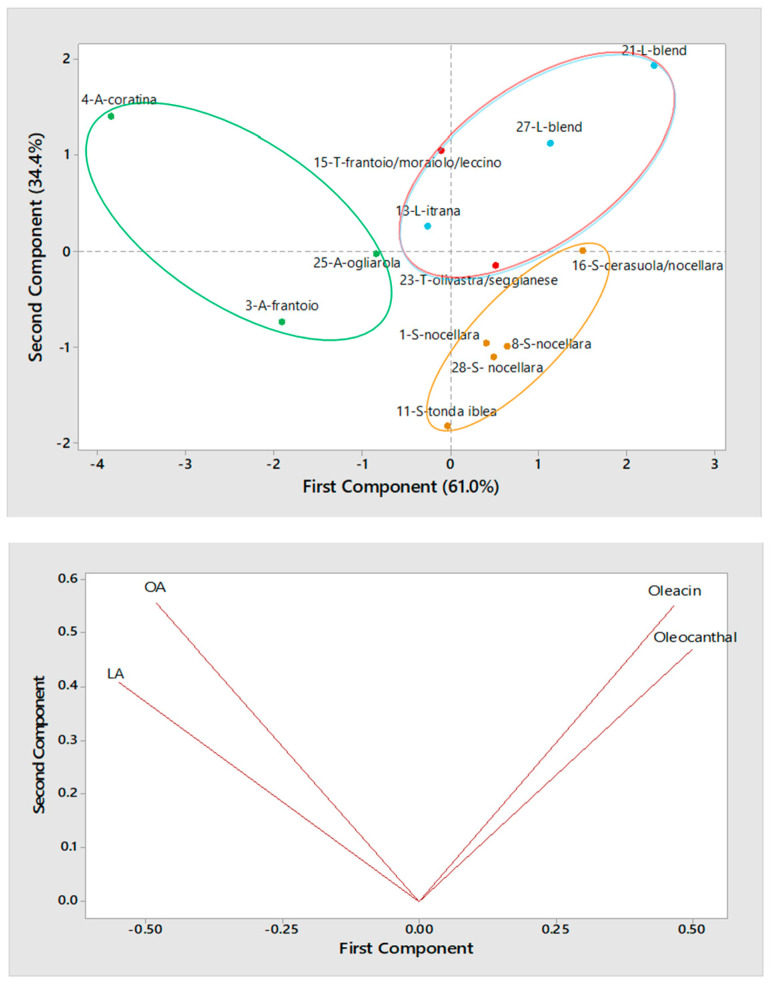
Score plot (**top panel**) and Loading plot (**bottom panel**) referred to the first two components provided by Principal Components Analysis (PCA) based on average normalized peak areas obtained for OA, LA, oleacin and oleocanthal (including the contribution of their carboxylic derivatives, if present) after the RPLC-ESI(-)-FTMS analysis of two replicated extracts for each of 13 Italian EVOOs produced during the 2016/2017 campaign using a three-phase decanter for horizontal centrifugation. Samples are labelled according to number (see Table 1), region (A = Apulia, L = Lazio, S = Sicily, T = Tuscany) and cultivar.

**Figure 6 molecules-26-00743-f006:**
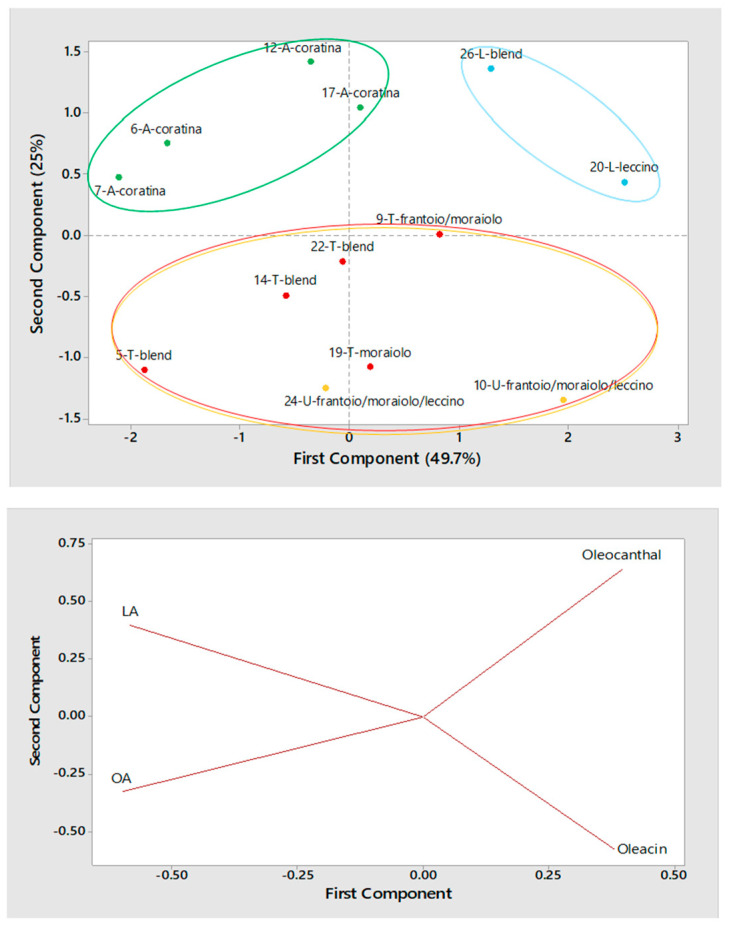
Score plot (**top panel**) and Loading plot (**bottom panel**) referred to the first two components provided by Principal Components Analysis (PCA) based on average normalized peak areas obtained for OA, LA, oleacin and oleocanthal (including the contribution of their carboxylic derivatives, if present) after the RPLC-ESI(-)-FTMS analysis of two replicated extracts for each of 13 Italian EVOOs produced during the 2016/2017 campaign using a two-phase decanter for horizontal centrifugation. Samples are labelled according to number (see Table 1), region (A = Apulia, L = Lazio, T = Tuscany, U = Umbria) and cultivar.

**Figure 7 molecules-26-00743-f007:**
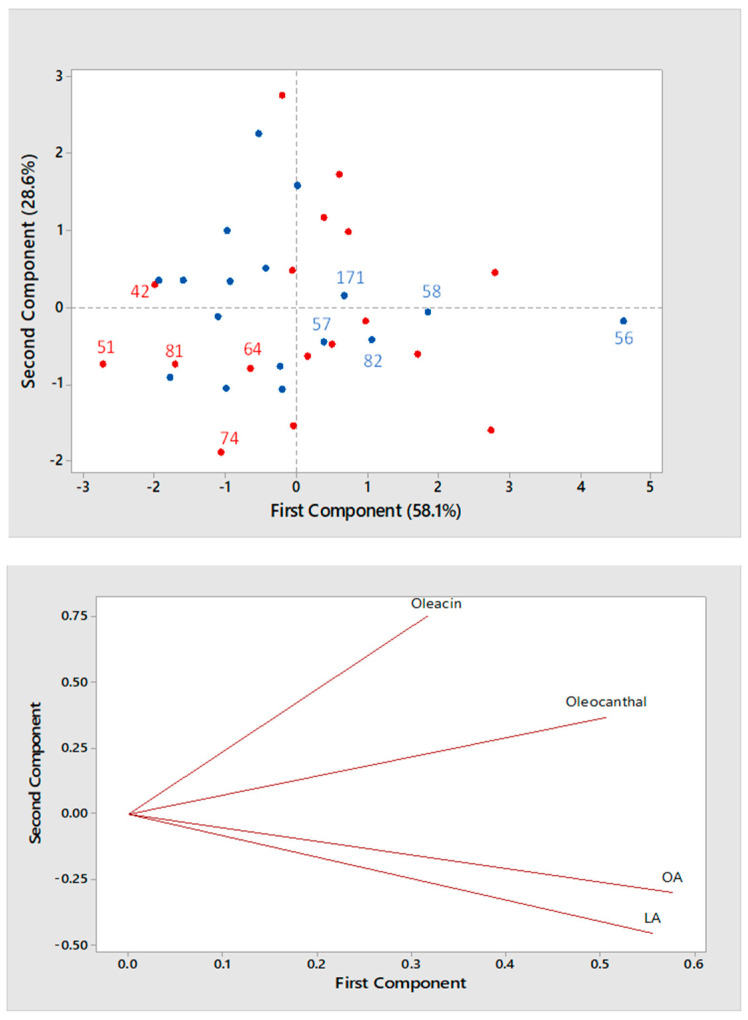
Score plot (**top panel**) and Loading plot (**bottom panel**) referred to the first two components provided by Principal Components Analysis (PCA) based on average normalized peak areas obtained for OA, LA, oleacin and oleocanthal (including the contribution of their carboxylic derivatives, if present) after the RPLC-ESI(-)-FTMS analysis of two replicated extracts for each of 34 Italian EVOOs produced during the 2017/2018 campaign. Olive oils produced using a three- or a two-phase decanter are indicated by blue and red dots, respectively. The sample numbering for emphasized samples is the same adopted in Table 1.

**Figure 8 molecules-26-00743-f008:**
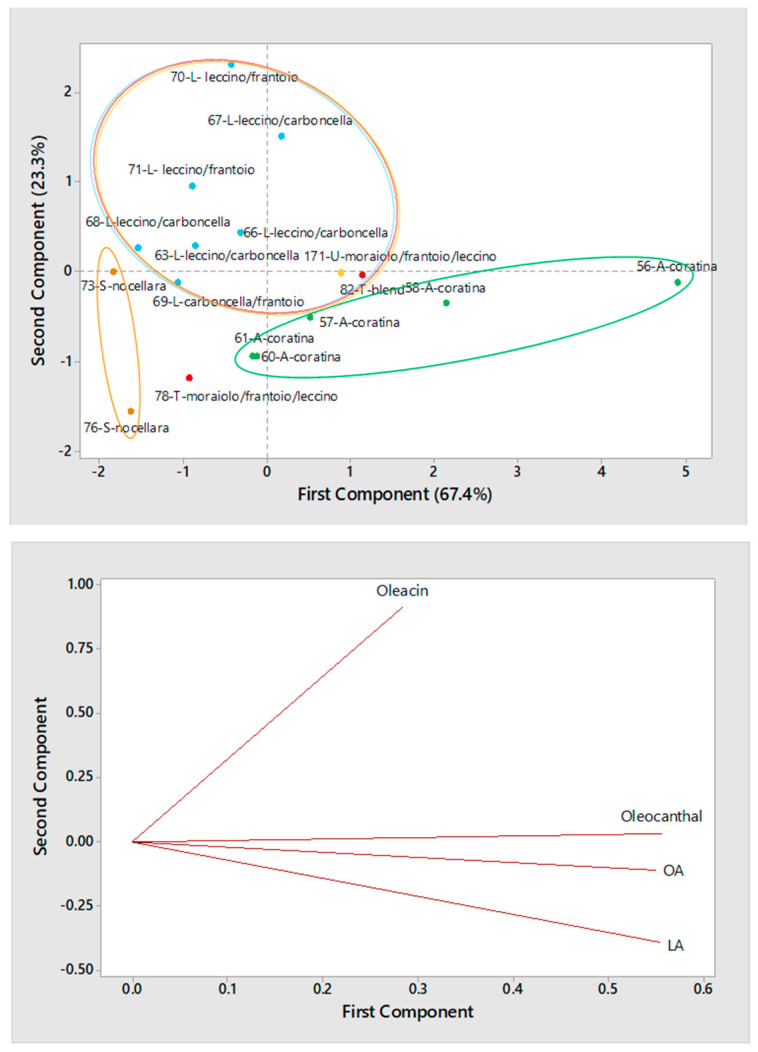
Score plot (**top panel**) and Loading plot (**bottom panel**) referred to the first two components provided by Principal Components Analysis (PCA) based on average normalized peak areas obtained for OA, LA, oleacin and oleocanthal (including the contribution of their carboxylic derivatives, if present) after the RPLC-ESI(-)-FTMS analysis of two replicated extracts for each of 17 Italian EVOOs produced during the 2017/2018 campaign using a three-phase decanter for horizontal centrifugation. Samples are labelled according to number (see Table 1), region (A = Apulia, L = Lazio, S = Sicily, T = Tuscany, U = Umbria) and cultivar.

**Figure 9 molecules-26-00743-f009:**
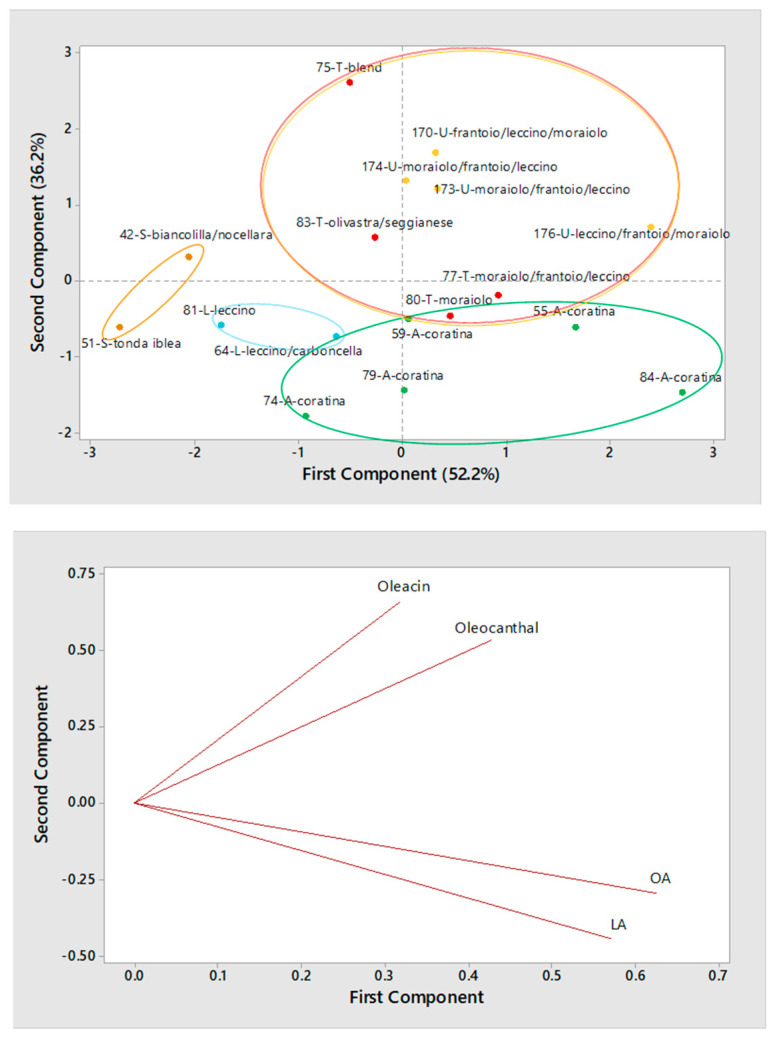
Score plot (**top panel**) and Loading plot (**bottom panel**) referred to the first two components provided by Principal Components Analysis (PCA) based on average normalized peak areas obtained for OA, LA, oleacin and oleocanthal (including the contribution of their carboxylic derivatives, if present) after the RPLC-ESI(-)-FTMS analysis of two replicated extracts for each of 17 Italian EVOOs produced during the 2017/2018 campaign using a two-phase decanter for horizontal centrifugation. Samples are labelled according to number (see Table 1), region (A = Apulia, L = Lazio, S = Sicily, T = Tuscany, U = Umbria) and cultivar.

**Table 1 molecules-26-00743-t001:** Summary of production-related information provided by producers for the 60 Italian olive oils analyzed in this work. Sample numeration is the same adopted in the context of VIOLIN research project. N/A = Not Available (information not provided by the producer).

**Campaign of Production: 2016/2017, 26 Extra-Virgin Olive Oils**					
**Olive Oil Sample #**	**Region**	**Olive Cultivar(s)**	**Horizontal** **Centrifugation**	**Type of Crusher**	**Malaxation** **Temperature/Time**
1	Sicily	Nocellara del Belice	Three-phase	Hammer	27 °C/60 min
3	Apulia	Frantoio	Three-phase	Hammer	N/A
4	Apulia	Coratina	Three-phase	Hammer	N/A
5	Tuscany	Blend	Two-phase	Blade	N/A
6	Apulia	Coratina	Two-phase	Hammer	21 °C/15 min
7	Apulia	Coratina	Two-phase	Hammer	27 °C/35 min
8	Sicily	Nocellara del Belice	Three-phase	Blade	26 °C/35 min
9	Tuscany	Blend	Two-phase	Blade	25 °C/10 min
10	Umbria	Frantoio, Leccino, Moraiolo	Two-phase	Hammer	22 °C/20 min
11	Sicily	Blend	Three-phase	Hammer	26 °C/60 min
12	Apulia	Coratina	Two-phase	N/A	N/A
13	Lazio	Itrana	Three-phase	Hammer	27 °C/30 min
14	Tuscany	Blend	Two-phase	Blade	N/A
15	Tuscany	Frantoio, Moraiolo, Leccino	Three-phase	Hammer	24 °C/30 min
16	Sicily	Cerasuola, Nocellara del Belice	Three-phase	Blade	27 °C/30 min
17	Apulia	Coratina	Two-phase	Hammer	25 °C/20 min
19	Tuscany	Moraiolo	Two-phase	Blade	27 °C/30 min
20	Lazio	Leccino	Two-phase	Hammer	23 °C/5 min
21	Lazio	Blend	Three-phase	N/A	N/A
22	Tuscany	Blend	Two-phase	Disk	N/A
23	Tuscany	Olivastra seggianese	Three-phase	N/A	N/A
24	Umbria	Frantoio, Moraiolo, Leccino, San Felice	Two-phase	Blade	21 °C/15 min
25	Apulia	Ogliarola	Three-phase	Disk	N/A
26	Lazio	Blend	Two-phase	N/A	N/A
27	Lazio	Blend	Three-phase	N/A	N/A
28	Sicily	Nocellara del Belice	Three-phase	Blade	26 °C/35 min
**Campaign of Production: 2017/2018, 34 Extra-Virgin Olive Oils**					
**Olive Oil Sample #**	**Region**	**Olive Cultivar(s)**	**Horizontal** **Centrifuge**	**Type of Crusher**	**Malaxation** **Temperature/Time**
42	Sicily	Biancolilla, Nocellara del Belice	Two-phase	Hammer	27 °C/25 min
51	Sicily	Tonda iblea	Two-phase	Disk- Hammer	27 °C/20 min
55	Apulia	Coratina	Two-phase	Hammer	N/A
56	Apulia	Coratina	Three-phase	Hammer	N/A
57	Apulia	Coratina	Three-phase	Hammer	N/A
58	Apulia	Coratina	Three-phase	Hammer	N/A
59	Apulia	Coratina	Two-phase	Hammer	N/A
60	Apulia	Coratina	Three-phase	Hammer	N/A
61	Apulia	Coratina	Three-phase	Hammer	27 °C/60 min
63	Lazio	Leccino, Carboncella	Three-phase	Disk	27 °C/45 min
64	Lazio	Leccino, Carboncella	Two-phase	Hammer	26 °C/35 min
66	Lazio	Blend	Three-phase	N/A	N/A
67	Lazio	Raja, Carboncella	Three-phase	N/A	N/A
68	Lazio	Leccino, Frantoio	Three-phase	N/A	N/A
69	Lazio	Carboncella, Frantoio, Salviana, Leccio del Corno	Three-phase	Hammer	N/A
70	Lazio	Leccino, Frantoio	Three-phase	N/A	N/A
71	Lazio	Leccino, Frantoio	Three-phase	N/A	N/A
73	Sicily	Nocellara del Belice	Three-phase	Hammer	27 °C/60 min
74	Apulia	Coratina	Two-phase	Hammer	21 °C/15 min
75	Tuscany	Blend	Two-phase	Blade	25 °C/15 min
76	Sicily	Nocellara del Belice	Three-phase	Hammer	26 °C/60 min
77	Tuscany	Moraiolo, Frantoio, Leccino	Two-phase	Blade	N/A
78	Tuscany	Moraiolo, Frantoio, Leccino	Three-phase	Hammer	24 °C/30 min
79	Apulia	Coratina	Two-phase	Hammer	25 °C/20 min
80	Tuscany	Moraiolo	Two-phase	Blade	27 °C/15 min
81	Lazio	Leccino	Two-phase	Hammer	23 °C/5 min
82	Tuscany	Blend	Three-phase	N/A	N/A
83	Tuscany	Olivastra seggianese	Two-phase	N/A	N/A
84	Apulia	Coratina	Two-phase	Hammer	27 °C/35 min
170	Umbria	Moraiolo, Frantoio, Leccino	Two-phase	Hammer	22 °C/20 min
171	Umbria	Moraiolo, Frantoio, Leccino	Three-phase	Hammer	24 °C/15 min
173	Umbria	Moraiolo, Frantoio, Leccino	Two-phase	Hammer	25 °C/90 min
174	Umbria	Moraiolo, Frantoio, Leccino	Two-phase	N/A	N/A
176	Umbria	Moraiolo, Frantoio, Leccino	Two-phase	N/A	N/A

## Data Availability

The data presented in this study are available on request from the corresponding author.

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
