# Peer review of "Bioactive Secoiridoids in Italian Extra-Virgin Olive Oils: Impact of Olive Plant Cultivars, Cultivation Regions and Processing"

_molecules, 2021, doi:10.3390/molecules26030743_

Round 1
Reviewer 1 Report
Manuscript Revision
Title: Bioactive Secoiridoids in Italian Extra-Virgin Olive Oils: Impact of Olive Plant Cultivars, Cultivation Regions and Processing.
Personally, I appreciate the effort made by the authors to study the variation in the concentration of phenolic compounds present in olive oil. I consider it of great interest to know what conditions allow to obtain a higher concentration of these compounds due to the proven bioactivities they possess. In the following lines, I will provide clear information to authors in order to improve the document.
In the introduction, I miss an explanation about how the study conditions (culture, region and processing) affect the chemical composition. It gives some results of which conditions are the best, but it is not explained why that happens. I think this point should carry more weight since it is what the article is about. On the other hand, when talking about the compounds under study, several names are cited for the same compound. I think that the IUPAC nomenclature would be better as a footnote in the figure since this would improve the understanding of the text. In such figure, it would be also interesting to put the chemical structure and the m / z and L conditions together. The way it is presented now, they appear to be two independent figures. If you want to leave it like this, then it would be convenient to make two figures.
In the results section, the geographical areas of study are explained. This would be interesting to explain it earlier in the introduction section. In section 2.2, the first paragraphs seem more method than not results. Figures of Score plot (top panel) and Loading plot (bottom panel) could be reorganized in order to use all the space.
I consider the discussion section to be very well written and explained. However, I miss some numerical support to explain the results. Section 4 is difficult to read. Also, it might be interesting to add an outline of the process.
At the end of the tables there is a map of the distribution of the oils. I think it would be more appropriate to put this information in the introduction. In addition, this image does not provide very relevant information either since these regions are unknown outside of Italy. That is why I believe that more work should be done to provide more information with this figure.
I believe that this manuscript needs a conclusion as the discussion is long and complex.
FINAL REMARKS
In my opinion, authors have carried out a really interesting study, with promising expectations for future research. The manuscript is clear and well written. I suggest MINOR REVISIONS. The study should be improved before publication.
Author Response
Preliminary note: the reply to each comment provided by the Reviewer is reported after a copy of the corresponding comment.
Comment: Personally, I appreciate the effort made by the authors to study the variation in the concentration of phenolic compounds present in olive oil. I consider it of great interest to know what conditions allow to obtain a higher concentration of these compounds due to the proven bioactivities they possess. In the following lines, I will provide clear information to authors in order to improve the document.
Reply: we thank the Reviewer for his/her positive comments on our paper. We have followed his/her suggestions to modify the manuscript and clarify several aspects.
Comment: In the introduction, I miss an explanation about how the study conditions (culture, region and processing) affect the chemical composition. It gives some results of which conditions are the best, but it is not explained why that happens. I think this point should carry more weight since it is what the article is about.
Reply: following the Reviewer suggestion, further details on the effects that different aspects, like olive cultivar and processing, may have on the secoiridoid content in olive oils have been added in the Introduction section.
Comment: On the other hand, when talking about the compounds under study, several names are cited for the same compound. I think that the IUPAC nomenclature would be better as a footnote in the figure since this would improve the understanding of the text. In such figure, it would be also interesting to put the chemical structure and the m / z and L conditions together. The way it is presented now, they appear to be two independent figures. If you want to leave it like this, then it would be convenient to make two figures.
Reply: we suppose that the Reviewer was referring to the names (Open Forms I, II, etc.) that we used to distinguish the different possible isoforms for each of the considered secoiridoids, when he/she referred to “several names”. We used that approach, both in this paper and in our previous works on olive oil secoiridoids, to emphasize the structural differences existing between those isoforms, being aware of the fact that diastereoisomers, geometrical isomers and enolic tautomers can be present for each of them. Those names were thus useful at least to group isoforms based on a common structural feature.
As for IUPAC names, they would change for each of the single isoforms belonging to the groups we indicated in new Figure 2, to take into account both the geometries of the C8-C9 or C8-C10 double bond, and all possible configurations for the stereogenic centres, excepting that for C5 (which is expected to have always the same configuration, being the original chiral centre of all those molecules).
Moreover, the numbering of atoms according to IUPAC rules will be different from that adopted in the figure, which is consistent with the numbering usually adopted in the literature concerning secoiridoids. This difference would generate some confusion in the readers, in our opinion.
For this reason, we prefer not to introduce IUPAC names. In any case, the IUPAC names for oleuropein (which has been corrected, since it was wrong in the previous version of the paper) and for elenolic acid have been reported in the paper.
As for former Figure 1, we think that it would be very difficult to report chemical structures for all the detected secoiridoid isoforms inside panels reporting XIC traces, due to the lack of space.
Consequently, following the Reviewer’s final suggestion, we have split former Figure 1 into two figures, namely, new Figure 2, reporting examples of XIC traces obtained for secoiridoids, and new Figure 3, showing molecular structures previously identified for the different isoforms of each secoiridoid, each labelled with the same numbers given to chromatographic peaks in new Figure 2.
Comment: In the results section, the geographical areas of study are explained. This would be interesting to explain it earlier in the introduction section.
Reply: the description of geographical areas of study has been moved into the Introduction and the map previously put below Table 1 was been put in the Introduction section as an independent figure (numbered as new Figure 1), as suggested by the Reviewer in one of his/her further comments (see below). Table 1 has been also moved at the end of the Introduction section, where it is cited for the first time.
Comment: In section 2.2, the first paragraphs seem more method than not results.
Reply: the first paragraph of section 2.2 was moved at the end of the introduction, in which more details were provided about the choice of Italian regions where olive oils were collected.
Comment: Figures of Score plot (top panel) and Loading plot (bottom panel) could be reorganized in order to use all the space.
Reply: we reduced the dimensions of the two panels included in each figure showing PCA results, in order to use the available space better. Basically, each of the figures was resized to about half a page. We preferred not to overlay the information related to the score and the loading plot in a unique plot, as it is sometimes made in papers showing PCA results, since this would have reduced the clarity of figures.
Comment: I consider the discussion section to be very well written and explained. However, I miss some numerical support to explain the results.
Reply: we are very grateful to the reviewer for his/her comments on the discussion section. Following the reviewer’s suggestion, we added and discussed 95% confidence intervals obtained for average normalized XIC peak areas of each of the four secoiridoids in EVOO produced with three or two-phase horizontal decanters in both the campaigns under study. The comparisons between those intervals reflected, on a numerical basis, the information previously obtained from PCA plots and enabled also an appreciation of the actual responses obtained for each of the four compounds of interest, which are correlated to their concentrations in the analysed EVOO extracts.
Comment: Section 4 is difficult to read. Also, it might be interesting to add an outline of the process.
Reply: we have modified section 4.4 to clarify the procedure adopted to obtain input values for PCA elaborations. An outline of the process has been added to the Supplementary Materials.
Comment: At the end of the tables there is a map of the distribution of the oils. I think it would be more appropriate to put this information in the introduction. In addition, this image does not provide very relevant information either since these regions are unknown outside of Italy. That is why I believe that more work should be done to provide more information with this figure.
Reply: the map and the relevant information has been moved to the Introduction section, as suggested. Further information has been added to explain the choice of regions where olive oils were collected.
Comment: I believe that this manuscript needs a conclusion as the discussion is long and complex.
Reply: following the Reviewer’s suggestion, a section with conclusions has been added at the end of the manuscript.
FINAL REMARKS
In my opinion, authors have carried out a really interesting study, with promising expectations for future research. The manuscript is clear and well written. I suggest MINOR REVISIONS. The study should be improved before publication.
Reply: we thank the Reviewer for his/her comments and for providing several useful suggestions to improve the manuscript.
Reviewer 2 Report
Well written and all results were explained properly.
The manuscript described well the analysis of Secoiridoids with help of reverse phase liquid chromatography coupled to electrospray ionization with high resolution Fourier transform mass spectrometry (RPLC-ESI-FTMS) depending on the plant cultivars, cultivation Regions and processing. They have compared the four major 181 secoiridoids in 60 Italian EVOO with help PCA" hor has reported a similar kind of study in their previous research work.
The strength of the manuscript is because the author included both data related to chemical analysis as well as the chemometric analysis while comparing the secoiridoids. Very few studies included both.
Experiments are explained well and a large number of samples has been included which make the results accurate rather than have less number of samples.
It would be interesting If they can provide the fragmentation for the secoiridoids
Author Response
Well written and all results were explained properly.
The manuscript described well the analysis of Secoiridoids with help of reverse phase liquid chromatography coupled to electrospray ionization with high resolution Fourier transform mass spectrometry (RPLC-ESI-FTMS) depending on the plant cultivars, cultivation Regions and processing. They have compared the four major 181 secoiridoids in 60 Italian EVOO with help PCA" hor has reported a similar kind of study in their previous research work.
The strength of the manuscript is because the author included both data related to chemical analysis as well as the chemometric analysis while comparing the secoiridoids. Very few studies included both.
Experiments are explained well and a large number of samples has been included which make the results accurate rather than have less number of samples.
It would be interesting If they can provide the fragmentation for the secoiridoids.
Reply: we are very grateful to the Reviewer for his/her so positive comments on our paper.
As for fragmentations, we prefer not to add information about them since the present paper had not a structural goal. Indeed, several detailed fragmentation schemes had been already reported, for all the secoiridoids of interest, in papers recently published by our research group, which are cited as Refs. 30, 31 and 32 in the present work. Those schemes were the basis for the structural assignments evidenced in Figure 1 (re-numbered as Figure 3 in the revised version of the paper).